**Data Availability Statement:** All relevant data are within the manuscript and its Supporting Information files.

**Funding:** This work was funded, in part, by the U.S. NIH/NIAID Collaborative Influenza Vaccine

# Hemagglutination Inhibition (HAI) antibody landscapes after vaccination with H7Nx virus like particles

**Hyesun Jang[1], Ted M. Ross**[1,2]*

**1** Center for Vaccines and Immunology, University of Georgia, Athens, GA, United States of America,
**2** Department of Infectious Diseases, University of Georgia, Athens, GA, United States of America

* tedross@uga.edu

## Abstract

### Background

A systemic evaluation of the antigenic differences of the H7 influenza hemagglutinin (HA) proteins, especially for the viruses isolated after 2016, are limited. The purpose of this study was to investigate the antigenic differences of major H7 strains with an ultimate aim to discover H7 HA proteins that can elicit protective receptor-binding antibodies against co-circulating H7 influenza strains.

### Method

A panel of eight H7 influenza strains were selected from 3,633 H7 HA amino acid sequences identified over the past two decades (2000–2018). The sequences were expressed on the surface of virus like particles (VLPs) and used to vaccinate C57BL/6 mice. Serum samples were collected and tested for hemagglutination-inhibition (HAI) activity. The vaccinated mice were challenged with lethal dose of H7N9 virus, A/Anhui/1/2013.

### Results

VLPs expressing the H7 HA antigens elicited broadly reactive antibodies each of the selected H7 HAs, except the A/Turkey/Italy/589/2000 (Italy/00) H7 HA. A putative glycosylation due to an A169T substitution in antigenic site B was identified as a unique antigenic profile of Italy/00. Introduction of the putative glycosylation site (H7 HA-A169T) significantly altered the antigenic profile of HA of the A/Anhui/1/2013 (H7N9) strain.

### Conclusion

This study identified key amino acid mutations that result in severe vaccine mismatches for future H7 epidemics. Future universal influenza vaccine candidates will need to focus on viral variants with these key mutations.

Innovation Centers (CIVICs) contract 75N93019C00052 and by the University of Georgia (UGA) (UGA-001). In addition, TMR is supported by the Georgia Research Alliance (GRA-001) as an Eminent Scholar.

**Competing interests:** The authors have declared that no competing interests exist.

## Introduction

Avian-origin influenza A hemagglutinin subtype 7 viruses (H7 AI viruses) circulate primarily in avian hosts. Humans are dead-end hosts for these virus infections and the H7 epidemics rarely persist among humans. However, some H7 influenza viruses may mutate in the human respiratory track and cause severe recurring epidemics [1]. There have been six epidemics caused by Asian H7N9 influenza viruses between 2013–2018 and this raises concern that this subtype may have the potential to cause influenza virus pandemics [2–4]. H7N2 influenza viruses caused epidemics in 2002 and 2003 and silently circulated among feline species and/or unknown reservoirs for fourteen years [5]. In the northeastern U.S., H7N2 influenza viruses have high affinity for the mammalian respiratory tract and are highly adapted to mammalian species with increased affinity toward α2–6 linked sialic acid [6]. In 2016, the feline H7N2 influenza viruses resulted the transmission from shelter cats to an attending veterinarian [7]. Even without adaptation, H7 influenza virus strains have caused at least five human epidemics since 2000: 1) the H7N1 influenza viruses infected people in Italy, 2) the H7N2 influenza viruses infected people in Northeastern U.S., 3) two distinct H7N3 influenza viruses infected people in North American and Eurasian countries, 4) one H7N4 infection case in China in 2018, and 5) people in Europe were infected with H7N7 influenza viruses [8]. These epidemics warrant that another avian influenza virus of the H7 subtype may infect and begin transmitting between humans to initiate the next H7 influenza virus pandemic.

For prompt production and distribution of vaccines during a pandemic emergency, the World Health Organization (WHO) has stockpiled candidate vaccine viruses (CVVs) for all H7 influenza viruses [9]. However, the antigenic differences of stockpiled CVVs have not been investigated, especially for the H7N9 viruses isolated after 2016 [10]. To prepare for the next H7 influenza virus epidemics, it is imperative to identify the antigenic differences of co-circulating H7 HA proteins and clarify the target coverage by the antigen.

There have been a small number of studies that investigated the antigenic differences of multiple H7 strains. Vaccination with divergent H7 HA immunogens isolated in 2009 from North American or Eurasian H7Nx viruses elicit immune responses that protect against Asian H7N9 influenza viruses [11]. Anti-H7 HA antiserum recovered from humans vaccinated with A/Anhui/1/13 H7 HA recombinant protein has broad binding activity to diverse H7 strains, including A/feline/New York/16-040082-1/2016 (H7N2) and to H7 HA from the A/turkey/Indiana/16-001403-1/2016 (H7N8) virus [12]. There were strong two-way cross-reactivity among H7N9, H7N2, H7N3 and H7N7 influenza viruses [13]. However, it is difficult to draw conclusions about the overall antigenic differences of co-circulating H7 influenza strains since each study used different representative reference strains and used antigens in different formats. In addition, these H7 HA antigens were isolated prior to 2016 and did not represent the current H7 HA variants. In this study, we aimed to investigate the antigenic differences of H7 influenza HA proteins that co-circulated in human over the last two decades.

## Materials and methods

### Study design

Overall study design was summarized in Fig 1 Briefly, genetic analyses was performed to select representative H7 strains between 2000 and 2020. Selected H7 HA sequences were expressed as virus like particles (VLPs) and subjected for the antigenic landscape analysis. Since it was not plausible to conduct cross-challenge studies across all eight viruses, cross-HAI assay was chosen for the antigenic landscaping. The HAI cut-off for protection was determined based on a mouse challenge study, which was described in prior to the cross-HAI titer analysis. A

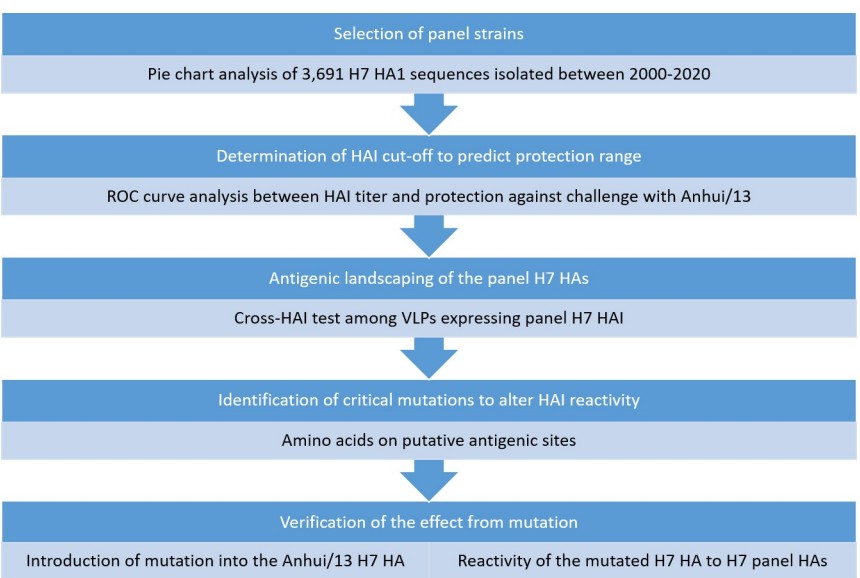

**Fig 1. Study design.** The eight representative strains were selected by pie chart analysis and subjected for antigenic landscaping using cross-HAI assay. Based on sequence comparison and antigenic landscaping data, critical mutation was identified and verified its role by site-directed mutation. *HAI*: *hemagglutination inhibitaion*, *ROC*: *receiver operating characteristic curve*, *VLPs*: *Virus Like Particles*.

mutagenesis study was followed to identify the critical mutation responsible for major antigenic changes.

## Alignment of HA amino acids sequences and virus like particle preparation

The H7 HA amino acid sequences uploaded on Global Initiative on Sharing All Influenza virus Data (GISAID) from 2000 to 2020 were downloaded. The sequences were aligned using Geneious software (Auckland, New Zealand). The amino acids 20–300 (HA1) region were extracted and partial or duplicate sequences were eliminated. The sequences were divided into three time periods/searches (2000–2012, 2013–2020 and 2013–2020 non-H7N9 sequences). The trimmed HA1 sequences of each group was aligned using the MUSCLE algorithm and clustered by 97% identity. Each cluster was illustrated as a pie chart using PRISM GraphPad Software (San Diego, CA, USA) and a panel of nine H7 strains of each cluster was selected.

Total of nine H7 HA sequences were expressed on the surface of virus like particles (VLPs), as previously described [14]. Briefly, the full-length H7 HA amino acid sequences were subjected to codon optimization for expression in a human cell line (Genewiz, Washington, DC, USA) and inserted into the pTR600 expression vector. The HEK 29T cells were transiently co-transfected (Lipofectamine™ 3000, Thermo Fisher Scientific, Waltham, MA USA) with plasmids expressing H7 HAs, HIV-1 Gag (optimized for expression in mammalian cells; Genewiz, Washington, DC, USA), and NA (A/Thailand/1(KAN-1)/2004 H5N1) (optimized for expression in mammalian cells; Genewiz, Washington, DC, USA). The cells (were incubated for 72 h at 37°C (Medigen Inc., Rockville, MD, USA). Supernatant was centrifuged in low speed and filtrated through a 0.22-μm sterile filter. Filtered supernatant was purified via ultracentrifugation (100,000 g through 20% glycerol, weight per volume) for 4h at 4°C. The pellets were subsequently resuspended in PBS (pH 7.2) and stored in single-use aliquots at 4°C until use.

The HA content of H7 VLPs was determined as previously described with slight modification [15]. Briefly, a high-affinity, 96-well flat bottom enzyme-linked immunosorbent assay

(ELISA) plate was coated with 5 to 10 μg of total protein of VLPs and serial dilutions of a recombinant H7 antigen (A/Anhui/1/2013 HA generated in house as previously described {Jang, 2020 #487}) in ELISA carbonate buffer (50 mM carbonate buffer, pH 9.5), and the plate was incubated overnight at 4˚C. The next morning, plates were washed in PBS with 0.05% Tween 20 (PBST), and then nonspecific epitopes were blocked with 1% bovine serum albumin (BSA) in PBST solution for 1 h at room temperature (RT). Buffer was removed, and then stalk-specific group 2 antibody (CR8020 {Tan, 2014 #488}) was added to the plate and incubated for two hours at 37˚C. Plates were washed and probed with goat anti-human IgG horseradish peroxidase-conjugated secondary antibody at a 1:3000 dilution and incubated for 2 h at 37˚C. Plates were washed 7 times with the wash buffer prior to development with 100 μL of 0.1% 2,2'-azino-bis(3-ethylbenzothiaozoline-6 –sulphonic acid; ABTS) solution with 0.05% $H_2O_2$ for 40 min at 37˚C. The reaction was terminated with 1% (w/v) sodium dodecyl sulfate (SDS). Colorimetric absorbance at 414 nm was measured using a PowerWaveXS (Biotek, Winooski, VT, USA) plate reader. Background was subtracted from negative wells. Linear regression standard curve analysis was performed using the known concentrations of recombinant standard antigen to estimate the HA content in VLP lots.

## Mouse study

C57BL/6 mice (*Mus musculus*, females, 6 to 8 weeks old) were purchased from Jackson Laboratory (Bar Harbor, ME, USA) and housed in microisolator units. The mice were allowed free access to food and water and were cared for under USDA guidelines for laboratory animals. All procedures were reviewed and approved by the Institutional Animal Care and Use Committee (IACUC). Mice (8 mice per group) were intramuscularly injected twice at four-week intervals with each VLPs (HA content = 3 μg) with AddaVax™ adjuvant (Invivogen, San Diego, CA, USA). Mice were bled at week 8. Mice were transferred to a biosafety level 3 (BSL-3) facility at the earliest availability (week 12), For viral challenge, mice were briefly anesthetized and infected with a 100 $LD_{50}$ dose of A/Anhui/1/2013 H7N9 via intranasal route ($1X10^3$ PFU/0.05 ml). At 4 days post-challenge, three mice in each group were randomly selected and sacrificed to harvest lung tissue. Remaining mice were monitored for the weight loss and euthanized at 14 days post-challenge. Weight loss more than 25% was used as a primary measurement for determination of humane endpoint. Also, dyspnea, lethargy, response to external stimuli and other respiratory distress was closely monitored for the determination of humane endpoint.

All procedures were in accordance with the NRC Guide for Care and Use of Laboratory Animals, the Animal Welfare act, and the CDC/NIH Biosafety and Microbiological and Biomedical Laboratories (IACUC number A2017 11-021-Y3-A11).

## Hemagglutination-Inhibition (HAI) assay

To evaluate the humoral response to each vaccination, blood was collected via submandibular bleeding using a lancet and transferred to a microfuge tube. Tubes were incubated at room temperature for at least 30 min prior to centrifugation, sera were collected and frozen at −20˚C ± 5˚C. A hemagglutination inhibition assay (HAI) assay was used to assess receptor-binding antibodies to the HA protein to inhibit agglutination of turkey red blood cells (TRBCs). The protocol is taken from the CDC laboratory influenza surveillance manual. To inactivate nonspecific inhibitors, mouse sera was treated with receptor destroying enzyme (RDE, Denka Seiken, Co., Japan) prior to being tested. Three parts of RDE was added to one-part sera and incubated overnight at 37˚C. The RDE was inactivated at 56˚C for 30 min; when cooled, 6 parts of sterile PBS was added to the sera and was kept at 4˚C until use. RDE treated sera was two-fold

serially diluted in v-bottom microtiter plates. Twenty-five μl of VLPs or virus at 8 HAU/50 μL was added to each well (4 HAU/25 μL). Plates were covered and incubated with virus for 20 min at room temperature before adding 0.8% TRBCs in PBS. The plates were mixed by agitation and covered; the RBCs were then allowed to settle for 30 min at room temperature. HAI titer was determined by the reciprocal dilution of the last well which contained non-agglutinated RBC. Negative (serum from naïve mouse) and positive serum controls (serum from H7 VLPs vaccinated mouse from previous study) were included for each plate. All mice were negative (HAI < 1:10) for pre-existing antibodies to currently circulating human influenza viruses prior to study onset.

## Plaque Forming Assay (PFA)

Viral titers were determined using a plaque forming assay using $1 \times 10^6$ Madin-Darby Canine Kidney (MDCK) cells, as previously described [16]. Briefly, lung samples collected at 4 days post challenge were snapped frozen and kept at −80˚C until processing. Lungs were serially diluted ($10^0$ to $10^5$) with sterilized phosphate buffered saline (PBS) and overlayed onto confluent MDCK cell layers for 1 h in 200 μl of DMEM supplemented with penicillin–streptomycin. Cells were washed after 1-hour incubation and DMEM was replaced with 3 mL of 1.2% Avicel (FMC BioPolymer; Philadelphia, PA)—MEM media supplemented with 1μg/mL TPCK-treated trypsin. After 48 h incubation at 37˚C with 5% CO2, the overlay was removed and washed 2x with sterile PBS, cells were fixed with 10% buffered formalin and stained for 15 mins with 1% crystal Violet. Cells were washed with tap water and allowed to dry. Plaques were counted and the plaque forming units calculated (PFU/mL).

## Determination of HAI cut-off to predict protection against challenge

The receiver operating characteristic (ROC) curve analysis between HAI titer and protection against Anhui/13 challenge, as previously described [17]. The ROC curve illustrates the diagnostic ability of a binary classifier system as its discrimination threshold is varied. To define protection as in binary format, we considered that individual mouse which maintained bodyweight between 90–100% of the original body weight during entire challenge study. The sensitivity and specificity of four cut-off values (VLP HAI titer = 40, 80,160, and 320) was analyzed for all each body weight cut-off. The sensitivity was calculated as "number of mice which showed hemagglutination inhibition (HAI) titer ≥ cut-off and was protected from the challenge study/ number of all protected mice". The Specificity was calculated as "number of mice which showed hemagglutination inhibition (HAI) titer < cut-off and unprotected from the challenge study/ number of all unprotected mice". The ROC curve was generated by connecting plots of sensitivity% versus 100-specificity% (false positive). The area under the curve (AUC) and Youden's index (Sensitivity + Specificity -1) was calculated by Prism (Graphpad software). The optimal cut-off was determined based on highest AUC or Youden's index to be used as a surrogate of protection.

## Site directed mutagenesis

The H7 HA numbering was based on a previous report [18]. The amino acids at residues 167 to 170 were changed from NAAF to NATF in the putative antigenic site B of A/Anhui/1/2013 H7N9 HA. The NATF amino acids are located at this position in the A/Turkey/Italy/589/2000 H7N1 HA molecules. By the single amino acid substitution, it is expected to introduce N-glycosylation site to the antigenic site B, located nearby the receptor binding site. The site directed mutagenesis was conducted with QuikChange II Site-Directed Mutagenesis Kit (Agilent, Santa Clara, CA, United States) in accordance with the manufactur's instructions. The Primer3

program (v. 0.4.0) was used to design mutagenesis primers. The plasmid was expressed as VLPs as described above. Expressed mutant VLPs were electrophoresed on a 10% Bris-Tris sodium dodecyl sulfate-polyacrylamide gel (SDS-PAGE) and stained by Comassie blue (Bio-rad, CA, USA)). The molecular weight for HA0 and HA1 estimated based on previous report {Alvarado-Facundo, 2016 #432} and online Peptide and Protein Molecular Weight Calculator (https://www.aatbio.com/tools/calculate-peptide-and-protein-molecular-weight-mw).

The Anhui/13 A169T H7 VLP was used to immunize eight C57B/L6 mice at day 0 and week 4. We measured the antigenic breath of the antisera collected at week 8. At week 8, all mice were challenged with Anhui/13 H7N9 wild type virus, as described above, and looked for weight loss, survival, and lung viral titer at 4 days-post-challenge.

### Statistical analysis

The difference in serum HAI titer and lung viral titer among groups was analyzed by ordinary one-way ANOVA, followed by Tukey's multiple comparison test. The difference in body weight loss of each time point was tested by Repeated Measures one-way ANOVA followed by Tukey's multiple comparison test. All statistical analysis was performed using Prism GraphPad Software.

## Results

### Phylogenetic analysis of H7Nx viruses isolated between 2000 and 2018

Among the 3,691 amino acid sequences uploaded to GISAID, almost half of the sequences (1740) showed 97% or higher HA1 amino acid similarity to A/Anhui/1/2013 H7N9 virus (Anhui/13-like). The uploaded amino acid sequences were biased to isolates from Asian H7N9 epidemics between 2013–2017. Since the Anhui/13-like sequences skewed the overall phylogenetic analysis, the sequences were separately aligned in three ways: 1) sequences isolated between 2000–2012 before the emergence of Asian H7N9 (Fig 2A), 2) H7N9 sequences isolated from 2013–2020 (Fig 2B), and non-H7N9 sequences isolated from 2013–2020 (Fig 2C).

Prior to the Asian H7N9 influenza virus outbreaks, the Eurasian and North American lineages represented the majority of H7 HA sequences in the database (53.14% and 45.95%, respectively) (Fig 2A). Interestingly, most of the Eurasian H7Nx influenza viruses isolated between 2000 to 2020, had high HA amino acid similarity (95% or more) to the oldest strain in our panel, A/Mallard/Netherland/12/2000 H7N3 (Table 1). Instead of a slow drift of HA1 amino acid sequences, genetic diversification of the H7Nx influenza viruses was driven by genetic reassortment that resulted in each cluster sharing unique neuraminidase subtypes (N1, N3, N7, N9). The North American lineage influenza viruses isolated between 2000–2012 were further subdivided into two distinct clusters that shared 92.5% amino acid similarity to each other (green and yellow segments in Fig 2A). During this 12-year period, the North American H7N3 influenza viruses had less genetic drift (<3%) and did not evolve into divergent subtypes (Teal pie in Fig 2A, 2B). The North American H7N2 influenza viruses spiked only in epidemics in early 2000s (2000–2003) and were not detected thereafter (yellow pie in Fig 2A).

The majority of viral sequences isolated from 2013–2020 were Anhui/13-like H7N9 influenza viruses (Fig 2B). Approximately 5.12% of the HA1 sequences had 3–5% difference in the amino acid sequence and represented as a separate clusters from Anhui/13-like HA sequences (Fig 2B). This small cluster of HA sequences consisted of the A/Guangdong/17SF003/2016 H7N9 (Guangdong/16)-like viruses, which evolved from Anhui/13 and clustered into a separate lineage in 2016–2017. Another separate phylogenetic cluster of Asian H7N9 viruses was the A/Shanghai/1/13 H7N9 (Shanghai/13)-like viruses. The Shanghai/13 was one of the earliest human H7N9 isolates in spring 2013, which evolved into a separate phylogenetic cluster from

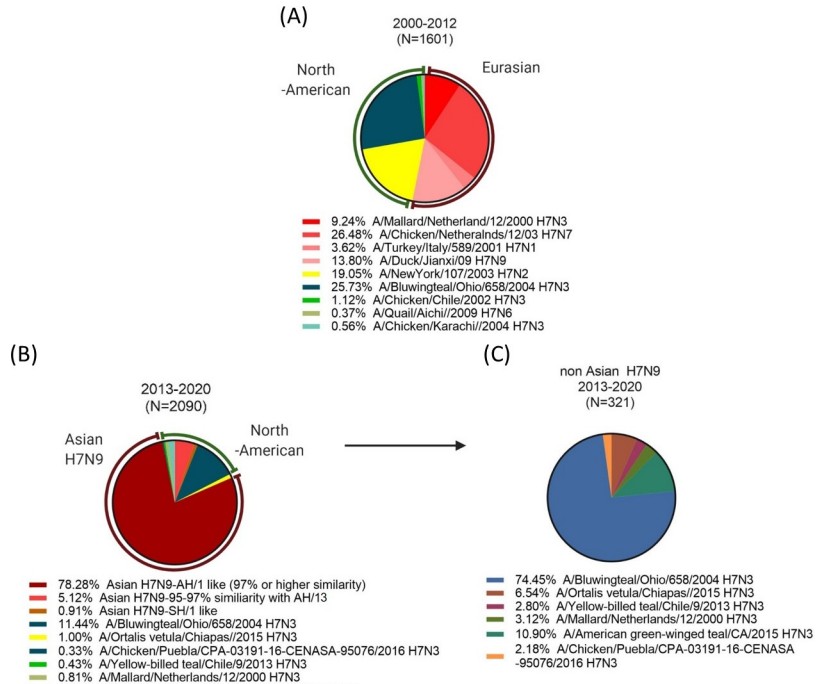

**Fig 2. Frequencies of influenza HA clusters in 2000 and 2018.** Total of 3633 Influenza HA sequences uploaded between 2000 and 2018 in GISAID databases were aligned to understand how the H7Nx viruses evolved. Due to the overwhelming number of Anhui/13-like viruses during Asian H7N9 epidemics, the pie chart analysis was separately conducted on sequences isolated before and after 2013 Asian H7N9 epidemics (A and B). The non-Asian H7N9 sequences isolated after 2013–2018 were further analyzed as a separate pie (C). The aligned sequences were clustered by 3% amino acid similarity and dissected into each pie. The viruses to represent each pie were chosen from WHO candidate vaccine viruses.

Anhui/13-like viruses [19, 20]. In this sequence analysis, the Shanghai/13 virus itself belonged to Anhui/13-like virus due to high homology (98.39% 9 AAs difference in HA1) of the HA amino acid sequences. However, the derivatives of Shanghai/13 had divergent sequences (<96% AAs homology, >17 AAs difference) to form a separate cluster that occupies ~1% of the overall HA sequences (Fig 2B).

The majority of non-Asian H7N9 influenza strain sequences uploaded on GSAID database between 2013 and 2020 were North-American H7N3 influenza virus derivatives, which represented ~26% of the HA amino acid sequences prior to the 2013 Asian H7N9 influenza virus outbreaks (Fig 2C). Most of the North American H7 influenza viruses were H7N3 viruses designated into four distinct HA sequence clusters. The A/American green-winged teal/CA/2015 H7N3 virus, which is the representative strain of the second largest cluster, is most likely derived from the H7N3 A/Bluewingteal/Ohio/658/2004 (Ohio/04) isolate. Interestingly, the northeastern U.S H7N2 strains have been rarely detected since 2004, except for one incident at an animal shelter in 2016 [7]. There are only 10 isolates that belong to the Eurasian lineage, but this is most likely due to the sampling bias for Asian H7N9 isolated in most Asian countries during that time period. All ten isolates had high homology to the NL/00 (H7N3) influenza virus.

## Selection of H7 panel strains

The panel of H7 influenza strains were selected to represent the antigenic diversity of H7Nx viruses during the last two decades. Asian H7N9 strains that are known to be antigenically

**Table 1. Selected panel strains.**

| Panel strains (full name) | GISAID Accession number | Amino acids homology (%) | | | | | | | |
|---|---|---|---|---|---|---|---|---|---|
| | | Shanghai/ 13 | Anhui/ 13 | Hunan/ 13 | Guangdong/ 16 | Italy/ 00 | Jiangxi/ 09 | Ohio/ 04 | New York/ 03 |
| **Shanghai/13** (A/Shanghai/1/2013 H7N9) | EPI744956 | - | | | | | | | |
| **Anhui/13** (A/Anhui/1/2013 H7N9) | EPI439507 | 98.39 | - | | | | | | |
| **Hunan/16** (A/Hunan/02650/2016 H7N9) | EPI961191 | 96.79 | 98.22 | - | | | | | |
| **Guangdong/16** (A/Guangdong/17SF003/ 2016 H7N9) | EPI919607 | 95.39 | 96.63 | 97.17 | - | | | | |
| **Italy/00** (A/Turkey/Italy/589/2000 H7N1) | EPI485603 | 95.54 | 95.54 | 94.83 | 93.09 | - | | | |
| **Jiangxi/09** (A/Duck/Jiangxi/3230/2009 H7N9) | EPI505699 | 96.43 | 96.43 | 95.01 | 93.62 | 97.14 | - | | |
| **Ohio/04** (A/Blue-wingeteal/Ohio/658/ 2004 H7N3) | EPI229595 | 84.82 | 84.82 | 83.76 | 83.51 | 84.82 | 86.07 | - | |
| **New York/03** (A/New York/107/2003 H7N2) | EPI141612 | 81.61 | 81.96 | 81.08 | 80.32 | 81.25 | 81.96 | 92.50 | - |

distinct from each other were selected [9]. For non-Asian H7N9 strains, three Eurasian strains and two North American strains were selected based upon remoteness in geography and time of isolation (Table 1 and Fig 3). The amino acid difference ranged between 1.61–5.14%, among Eurasian strains despite of dispersed isolation and time points of collection. The North American strains shared ~81–86% amino acid homology with Eurasian strains. Even though the Ohio/04 and New York/03 strains were isolated within a year from geographically similar regions, they shared 92.5% of the same HA amino acids. It was interesting that only few of mutations were observed from the putative antigenic site of nine strains isolated during two decades (Table 2). Of note, the hallmark mutation that causes N-linked glycosylation in antigenic site B was observed from Italy/00 (Table 2, blue-color coded and asterisk).

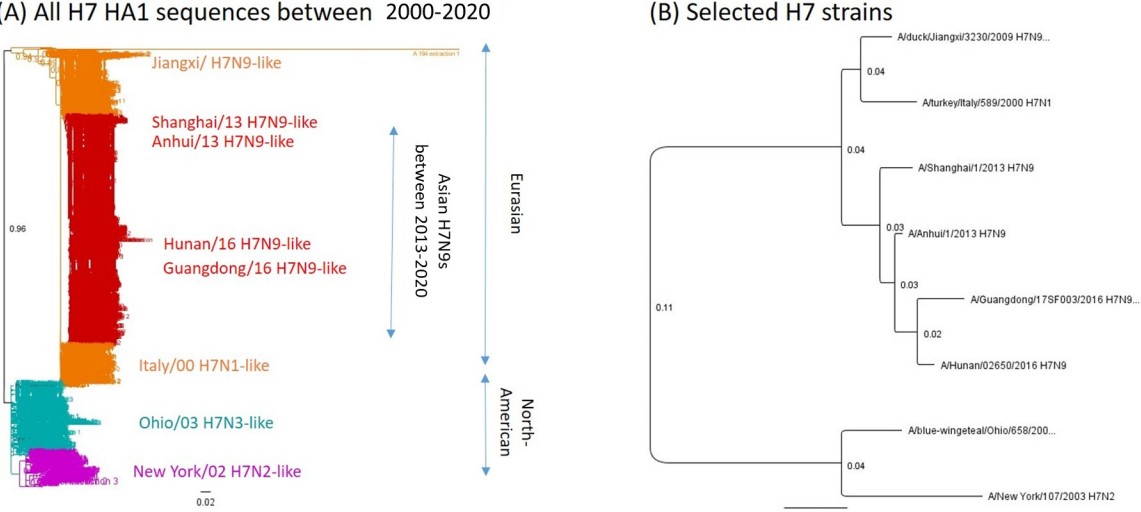

**Fig 3. Phylogenetic relations among selected H7 strains.** The phylogram based on HA1 amino acid sequences (H7 HA$_{20-300}$) was constructed by Neighbor-Joining method with the boot-strap resampling (100 replicates) using the Geneious software (Auckland, New Zealand). The horizontal branch lengths are proportional to the number of nucleotide changes. **(A) Phylogeny of all H7 HA1 amino acid sequences** Red: Asian H7N9s isolated between 2013–2020, Orange: Eurasian H7NXs isolated between 2000–2012, Cyan: Ohio/03 H7N3-like cluster, Magenta: New York/02-like cluster **(B) Selected panel strains** Phylogenetic trees based on HA1 amino acid sequences of selected H7 panel strains.

## Determination of HAI cut-off for protection

Mice were vaccinated with virus-like particles expressing the panel H7 HA sequences and challenged with Anhui/13 H7N9 virus. This challenge study was conducted to determine HAI cut-off for protection. All vaccinated mice had high titer antibodies with HAI activity to the Anhui/13 H7N9 virus except those vaccinated with the NY/02 virus (Fig 4A). The HAI titer against live Anhui/13 virus showed similar pattern, albiet with lower titers (Fig 4B). The level of cross-HAI reactivity did not directly correlate with the antigenic similarity (Table 1 and Fig 4).

Following challenge with Anhui/13, mice were observed for clinical signs and mortality (Fig 5). To determine the protection, average body weight loss 5% or less was considered as minimal body weight loss (Dotted line in Fig 5A). Mock vaccinated mice lost greater than 15% body weight by day 7 post-infection, which was similar to mice vaccinated with NY/02 VLPs (Fig 5A) with 60% of the mice reaching clinical endpoints and were sacrificed (Fig 5B). Mice vaccinated with Jiangxi/09 or Guangdong/16 lost 12% body weight. Mice vaccinated with the other VLPs lost between 5–8% body weights, except for mice vaccinated with Hunan/16 that maintained their average body for the entire challenge period. Most mice survived challenge (Fig 5B). One mouse died in the Jiangxi/09 group and 2 mice died in the Guangdong /16 group. Little to no virus was detectable in the lungs of mice vaccinated with Anhui/13 or Shanghai/13, and only one mouse in the Hunan/16 group had detectable virus (Fig 5C).

The ROC curve analysis was conducted between HAI titer and protection data following Anhui/13 challenge study (S1–S4 Tables). Protection to the Anhui/13 H7N9 challenge was determined if individual could maintain % body weight between 90% and 100%. The selection of the cut-off was determined by two criteria: maximizing sensitivity (AUC of the curve) and maximized the summation of sensitivity and specificity (Youden's index) [21]. As a representative data for ROC analysis, S1 Fig. Illustrated the ROC curve of each HAI cut-off to predict protection defined as 5% or less body weight loss. The highest sensitivity of the prediction was observed as the maximum area under the curve when the VLP HAI cut-off was 1:80 (S1B Fig). The Youden's index (specificity + sensitivity -1) was highest when the HAI cut-off was 1:160 (S1C Fig). Thus, we used the range 1:80 as the cut-off of HAI titer that can provide protection against a stringent challenge by each H7 influenza virus in panel. The absolute protection is expected if the VLP HAI titer is higher than 160, while HAI titer between 80–160 is expected to provide marginal protection. When applying the cut-offs determined by the ROC analyses, the pre-challenge HAI titer appears to correctly predict the level of protection in weight loss (Figs 4A and 5A) in a stringent Anhui/13 challenge.

**Table 2. Putative antigenic sites of selected panel strains.**

| Putative antigenic site | | A | B | E | B | D | E | C |
|---|---|---|---|---|---|---|---|---|
| H7 numbering | | 148–153 | 160–169 | 179–183 | 197–206 | 213–229 | 268–273 | 284–295 |
| H7 Strains | Anhui/13 | RRSGSS | WLLSNTDNAA | NTRKS | TAEQTKLYGS | VGSSNYQQSFVPSPGAR | FLRGKS | ANCEGDC |
| | Shanghai/13 | RRSGSS | WLLSNTDNAA | NTRK**N** | TAEQTKLYGS | VGSSNYQQSFVPSPGAR | FLRGKS | A**D**CEGDC |
| | Hunan/16 | **K**RSGSS | WLLSNTDNAA | NTRKS | TAEQTKLYGS | VGSSNYQQSFVPSPGAR | FLRGKS | ANCEGDC |
| | Guangdong/16 | RRSGSS | WLLSNTDNAA | NTK**E**S | TAEQTKLYGS | VGSSNYQQSFVPSPGAR | FLRGKS | ANCEGDC |
| | Jianxi/09 | RRSGSS | WLLSNTDNAA | NTRK**D** | T**T**EQTKLYGS | VGSSNYQQSFVPSPGAR | FLRGKS | ANCEGDC |
| | Italy/00 | **K**RSGSS | WLLSNTDNA**T**\* | NTRK**D** | **N**T**EQTKLYGS | **I**GSSNYQQSFVPSPGAR | FLRGKS | ANCEGDC |
| | Ohio/04 | RRSGSS | WLLSN**S**DNAA | N**PR**NK | **AT**EQTKLYGS | VGSS**K**YQQSF**T**PSPGAR | F**F**RGE**S** | **SG**CEGDC |
| | New York/03 | **T**RSGSS | WLLSN**S**DNAA | N**PR**NK | **VS**EQTKLYGS | V**R**SS**K**YQQSF**T**P**N**PGAR | F**F**RGE**S** | **SSCR**GDC |

Difference in the amino acid from Anhui/13 was color-coded as blue.

\*Hall mark mutation which can cause N-glycosylation.

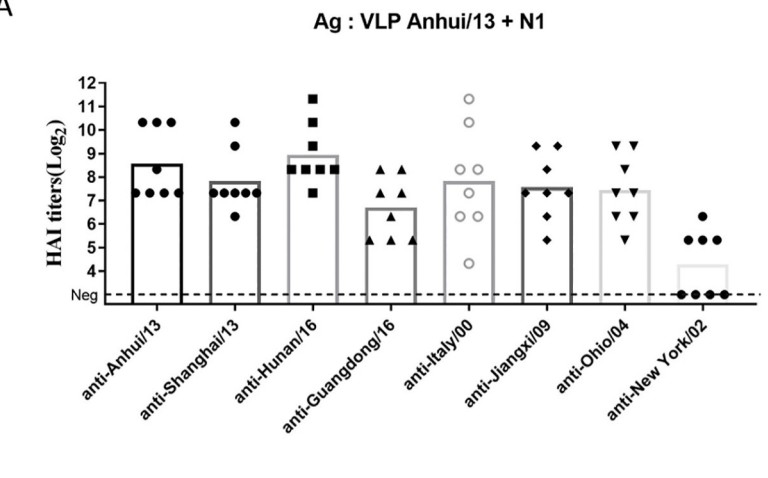

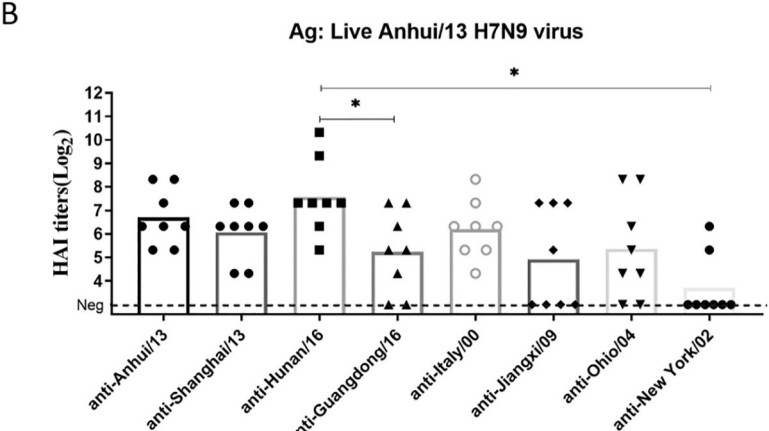

**Fig 4. HAI antibody titers against Anhui/13 H7 HA in mouse sera after immunization with various H7 HAs.**
Serum samples collected at week 8 were tested for the HAI antibody response specific to A/Anhui/1/2013 H7 virus like
particles (VLPs) and A/Anhui/1/2013 H7N9 virus (A and B, respectively. Individual titer was plotted and the mean
value was presented as bars. Dotted line represents the lower detection limit (10 HAI units). *p<0.05, **p<0.01.

## Cross-reactiveness amongst all H7 panel strains

For a comparison of cross-reactive HAI activity, the cut-off 80 was also applied. The HAI anti-
bodies elicited by each H7N9 VLPs had a broad-range of cross-reactive antibodies (Fig 6). The

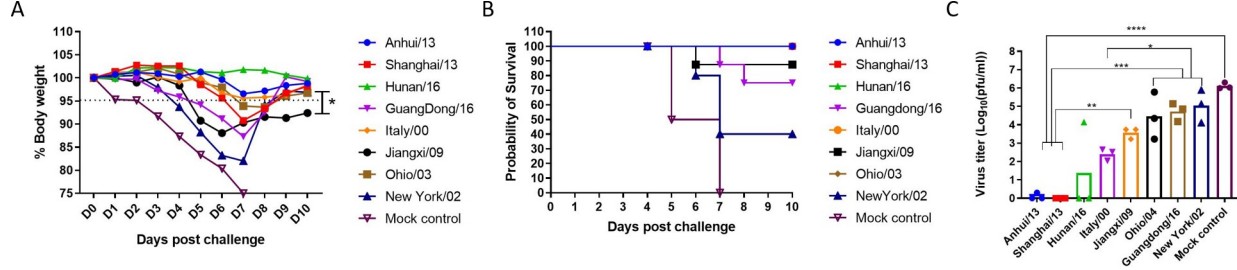

**Fig 5. Protection against stringent H7N9 challenge.** C57BL/6 mice (8 mice/group) vaccinated with H7 VLPs at week 0 and 4 were intranasally
infected with the A/Anhui/1/2013 H7N9) virus. Mice were monitored daily for weight loss (A and B, respectively) and viral lung titers in selected
mice at day 4 post infection (C). Weight loss and lung viral titer was presented as mean± standard deviation (A and C). *p<0.05, **p<0.01,
***p<0.001, ****p<0.0001.

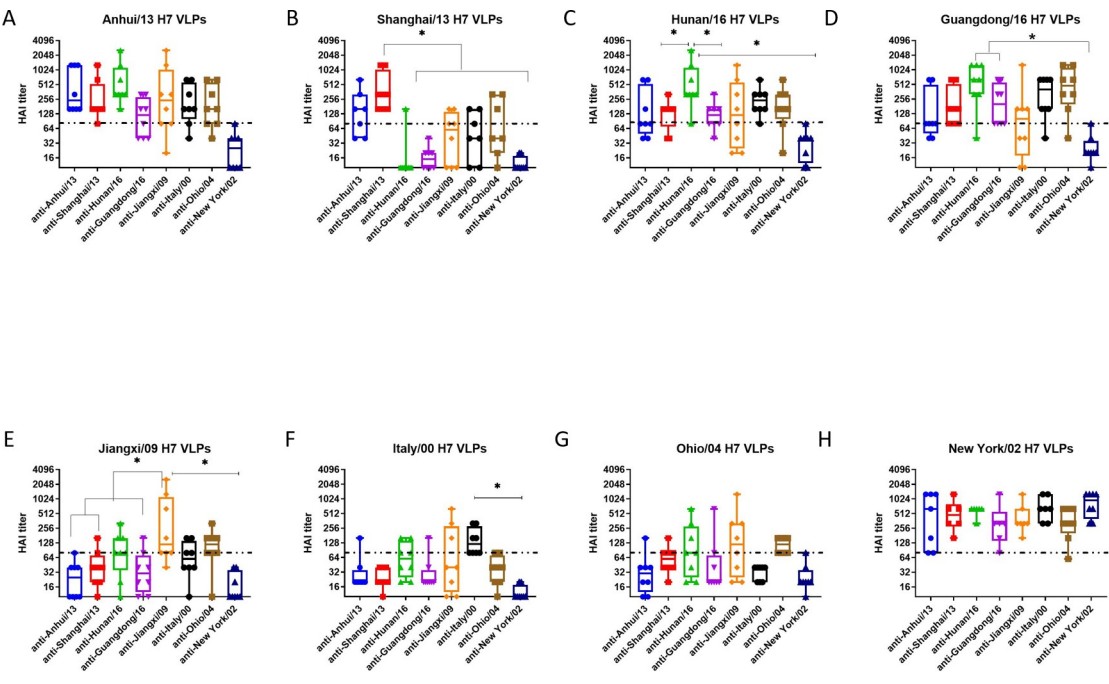

**Fig 6. Cross-reactiveness among H7 panel strains.** The week 8 sera was tested for the cross-reactivity to H7 VLPs expressing HA from all eight panel strains. Individual titer was plotted. Interquartile range, median, minimum and maximum values were presented as box, middle line, upper and lower whiskers, respectively. Dotted line indicates the cut-off for the protection (80 HAI unit). *p<0.05, **p<0.01.

cross-reactivity of each antisera did not correlate with the amino acid sequence similarity of the HA (Table 1 and Fig 6). Mice vaccinated with the four Asian H7N9 strains (Anhui/13, Shanghai/13, Guangdong/16, and Hunan/16) had cross-reactivity to each other (Fig 6A–6D), but did not recognize Jiangxi/09, Italy/00 or Ohio/04 (Fig 6E–6G). Antisera to the Jiangxi/09 or Ohio/04 showed broad cross-reactive HAI activity against all the H7 viruses in the panel, except to Italy/00 (Fig 6). In contrast, anti-Italy/00 sera had broad HAI activity against all the viruses in the panel, except against Jiangxi/09 and Ohio/00 (Fig 6). Mice vaccinated with NY/02 VLPs elicited antibodies with HAI activity against the homologous NY/02 virus, but did not recognize any of the other H7 viruses (Fig 6).

## Influence of glycosylation site

With regard to the unique antigenic profile of Italy/00, we found that there was a putative glycosylation site at $HA_{169}$ (H7 numbering from our own sequence alignment) (Table 2). Since the location of putative *N*-linked glycosylation was located in antigenic site B, we hypothesized that glycosylation at this location may be responsible for the unique antigenic profile of Italy/00. To test the hypothesis, we introduced a mutation into the HA nucleotide sequence of Anhui/13 (HA A169T) (Fig 7A) and looked for the change in reactivity elicited antisera by each VLP vaccine (Fig 7B). Interestingly, the reactivity of VLP expressing the Anhui/13 HA A169T mutation elicited antibodies with a significant decrease in HAI activity against Anhui/13 and Hunan/13, but no change against the other 6 viruses (Fig 7B). According to the predicted trimeric structure (Protein data base number = 4N5J), the glycosylation site appear to be located on the antigenic site B, and next to the receptor binding site (Fig 7C). The VLPs expressing WT-, and A169T- Anhui/13 H7 HAs were characterized by Comassie blue stained

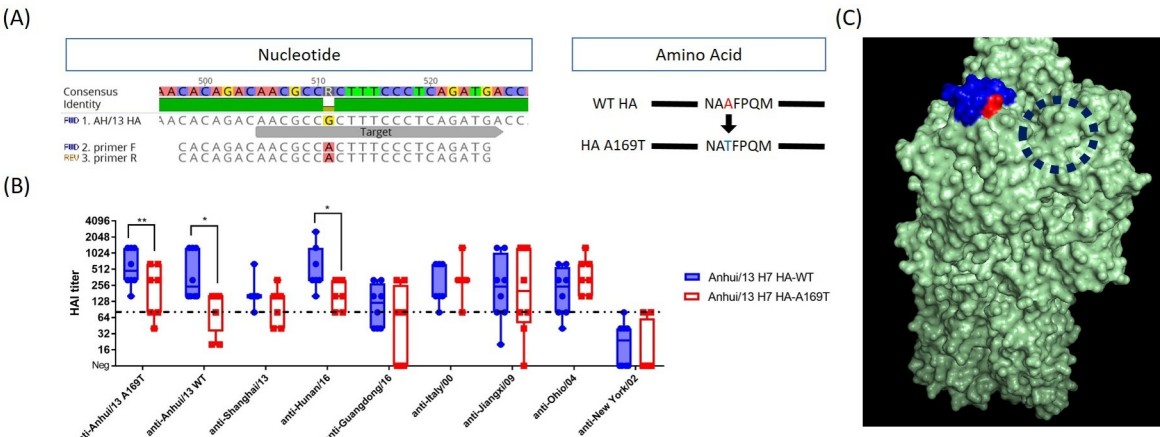

**Fig 7. Alanine to threonine mutation at HA169 resulted in significant antigenic change in Anhui/13 H7 HA. (A) Mutagenesis to Anhui/13 H7 HA** Site directed mutation was conducted on plasmid expressing wildtype (WT) Anhui/13 HA. The mutation is expected to result in alanine to threonine substitution at HA169 (H7 numbering), primer F = forward primer, primer R = reverse primer **(B) Change of cross-reactiveness by the mutagenesis** The plasmid with mutation was expressed as virus-like particle (VLP) tested for the reactivity to anti-H7 panel sera. Individual HAI titer was plotted. The box indicates the mean± standard deviation. *p<0.05, **<0.1, dotted line: protection cut-off (80 HAI unit). **(C) Predictive location of mutation on the HA trimer** The trimeric structure of Anhui/13 H7 HA was generated using the 3D-JIGSAW algorithm, and renderings were performed using MacPyMol. The trimeric structure was based on the structure on protein data base (PDB number = 4N5J). The putative antigenic site B and mutation site (H7 HA169) was shown in blue and red, respectively. Dashed circle indicates receptor binding site.

SDS-PAGE with or without PNGaseF treatment (S3 Fig). In presence of the PNGaseF, which removes the N-linked glycans, the HA0 and HA1 band or both WT and A169T VLPs was observed at similar level (left two lanes). But without the PNGaseF treatment, the HA0 band for A149T VLPs (red arrow head in S3 Fig) showed slightly higher molecular weight than WT VLPs, which suggests the addition of glycosylation to the mutant VLPs.

We also immunized C57B/L6 mice with the Anhui/13 A169T VLPs and looked for the anti-genic breath of the antisera and protection efficacy against Anhui/13 WT H7N9 challenge (Fig 8). Interestingly, the HAI titer to the Anhui/13 A169T VLPs (homologous antigen) was signifi-cantly lower and showed bigger standard deviation than the HAI titer to the Anhui/13 WT (Figs 7C and 8A). The HAI activity to the Shanghai/13 VLPs was similar with the titer to the Anhui/13 A169T VLPs (Fig 8A). High reactivity to the New York/02 VLPs (Fig 8C), which was also observed from other antisera for all 8 panel strains (Fig 6). The HAI reactivity to the Hunan/16, Guangdong/16, Jiangxi/09, Italy/00, and Ohio/03 H7 VLPs was significantly lower than the titer to the Anhui/13 WT and New York/02 H7 VLPs. In consistent with the high HAI titer to the Anhui/13 WT H7 VLPs, the mice were completely protected from weight loss and onset of any clinical symptom by the lethal challenge with the Anhui/13 WT H7N9 virus (Fig 8C and 8D). There was no detectable infectious viral titer in the lung collected at day 4 post challenge, which was clearly contrasted with the naïve control mouse (Fig 8B).

## Discussion

This study investigated the antigenic differences of selected H7 panel influenza HA proteins. Since most available H7 HA sequences originated from major human infections, the selected H7 panel strains were similar with the list of candidate vaccine viruses (CVVs) from the WHO [10]. There was a high similarity of amino acid sequences in the putative HA antigenic sites (Table 2). In addition, antibodies elicited by these HA antigens had HAI activity to most of these H7 viruses (Fig 6). It was consistent with previous findings showing that broad cross-

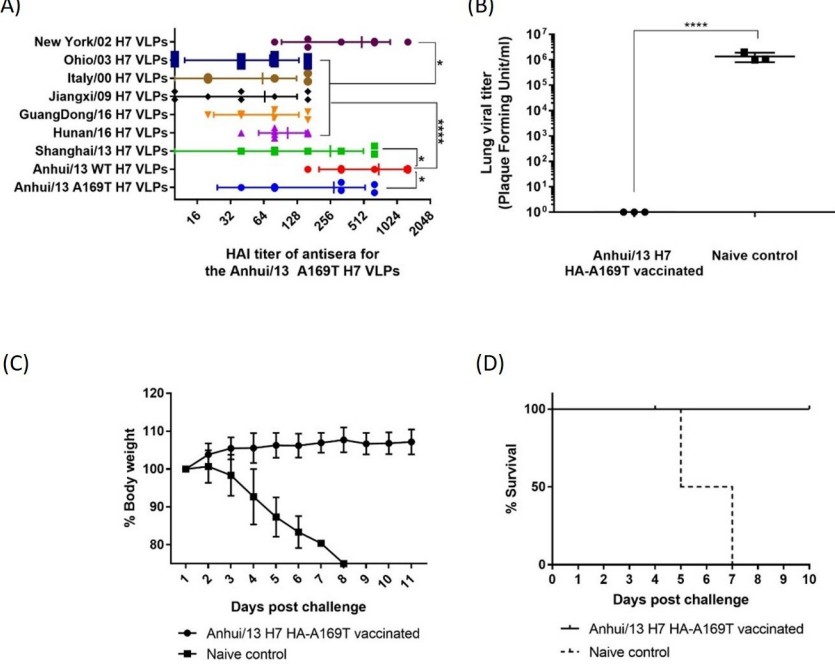

**Fig 8. Immunization with Anhui/13 A169T H7 VLP.** C57BL/6 mice (8 mice) was vaccinated with Anhui/169T H7 VLPs at week 0 and 4. Vaccinated mice were bled and the serum was tested for the antigenic breath across panel H7 strains (A). At week 8, vaccinated mice were intranasally infected with 10e+5 PFU of the A/Anhui/1/2013 H7N9) virus. Mice were monitored daily for weight loss and survival (C and D, respectively) and viral lung titers in selected mice at day 4 post infection (B). Weight loss and lung viral titer was presented as mean± standard deviation (A&C). *p<0.05, ****p<0.0001.

reactivity among H7 influenza viruses isolated from both North American and Eurasian countries [12, 22].

Before this study, Joseph et al conducted similar study with ten H7 influenza viruses isolated between 1971 and 2004 [23]. The selection of panel strains was based on phylogenetic relations and geographic locations. The cross-reactive neutralizing antibody response was observed similar with our study. For example, despite of phylogenetic heterogenicity, the antisera for two H7N3 viruses isolated from American and Eurasian countries (A/chicken/Chile/4322/02 (H7N3) and A/turkey/England/63 (H7N3), respectively) were cross reactive each other. The antisera for A/turkey/VA/55/02 (H7N2) was poorly cross-reactive to other H7 viruses, while the H7N2 antigen could be recognized by other antisera. Our study extended the analyses into more recent H7 strains, and identified a major mutation which could significantly alter the antigenic profile.

From both our study and the work of Joseph et al, the H7N2 viruses isolated from northeastern U.S. in early 2000 showed unique antigenic profile. In phylogenetic analysis, the H7N2 viruses were uniquely clustered from other H7 viruses due to the large truncation at the putative receptor binding site (H7 HA) (S2 Fig). The unique structure of HA appear to ease the binding of antibodies from other antisera, while the antisera for the H7N2 was lack of major epitope. Meanwhile, The HAI titer against Italy/00 and Ohio/04 VLPs was observed low from all antisera, even to the homologous antisera. Only anti-Italy/00 antibodies against Italy/00 VLP were above the cut-off, and only anti-Ohio/04 antibodies against Ohio/04 VLPs were above cut-offs. It seemed that in comparison to other VLPs, the access to the two VLPs were much restricted. The presence of glycosylation on the receptor binding site also significantly

impair the reactivity to the homologous antisera; even the antisera collected from mice vaccinated with the Anhui/13 A169T H7 VLPs detected the Anhui/13 WT H7 VLPs better (Fig 8A). We can explain that the Italy/00 has glycosylation site near the receptor binding site, so even homologous antisera showed relatively lower access to the VLP. We could not find plausible explanation for the Ohio/04 VLPs, but suspect that the structure of Ohio/04 expressing VLP might have hindered the access of the antibodies.

The level of cross-HAI activity among H7 HA proteins did not follow phylogenetic similarity or geographic origin. Instead, mutations that altered the glycosylation pattern around the receptor binding site (RBS) played a critical role in shaping the antigenic profile. A single amino acid substitution (HA A169T) caused a significantly reduce the reactivity to antisera specific for Asian H7N9 strains. The mutation did not significantly influence on reactivity to other anti-sera, which suggests that such antigenic site was not dominant recognition site by such antibodies. The mutations were based on the distinctive antigenic profile of Italy/00 H7 HA. This protein has an N-linked glycosylation site (NATF) at residue 167–170 of the HA molecule (Table 2). The putative location of the *N*-glycosylation is adjacent to the receptor binding site of the trimeric form of HAs (Fig 6C). Spontaneous occurrence of the N-linked glycosylation sites at the same location in H7 HA proteins was previously reported during the H7N1 epidemics in Italy in the early 2000's [24]. The study used reverse genetics to generate virus which has the corresponding mutation A149T (A169T by our numbering) and showed that the single mutation alone resulted in glycosylation by electrophoresis [24]. Also, the mutation was spontaneous and stable during the passage of the H7N1 viruses in turkeys, which suggests that the mutation can naturally occur during circulation in poultry species [24]. There was no significant influence of the glycosylation site on host tropism, however, the potential change in antigenicity was not investigated [24]. The latest study published in 2020 also verified that the corresponding mutation A151T (A169T by our numbering) occurred in one of the escaping mutants and proved that the mutation results in glycosylation [25]. But both studies did not investigate its influence on cross-reactivity to other H7 strains. The closest finding to our study was a study conducted by Zost that demonstrated a lysine to threonine mutation at residue 170 of H3 HA (corresponding to H7 HA169) resulted in a significant change in the glycosylation pattern at antigenic site B and antigenic mismatch to the parental virus [26]. This was not limited to residue 169, the glycosylation at a separate location (H7 HA 141T), which also naturally occurs, hindered the access of the epitope to neutralizing antibodies [18]. This motif was initially found at seven amino acids upstream to antigenic site A in the A/Netherlands/219/2003 H7 HA [18]. Similar to this study, introduction of the corresponding mutation into the A/Shanghai/2/2013 H7 HA (identical HA sequence of Anhui/13) decreased the binding of specific monoclonal antibodies and facilitated HA-mediated entry of the virus [18]. Our study identified that single amino mutation could significantly reduce the reactivity to the homologous strains, and it seems that there could be more signature mutations on H7 HAs, which can results in vaccine mismatch. H7 HA vaccine strategies should aim to identify more of such mutations and to cover such variants to prevent severe vaccine mismatches.

Serum HAI assay has been known to be best surrogate for protection {Dunning, 2016 #484}. Since the human challenge study conducted in the 1970s, the 1:40 HAI titer has been used to predict vaccine effectiveness when an appropriate challenge study is not plausible, such as the annual flu vaccine approval process [27–29]. While the 1:40 1:40 HAI titer cut-off is sufficient to provide a rough prediction, the specificity of this prediction can be improved by increasing the HAI titer cut-off [28, 30]. This is particularly true for subjects with higher revaccination risks, such as the elderly population [28, 30]. Also, the cut-off should be optimized based on the format of testing antigen. The VLPs expresses same HA amino acid sequences with wild type viruses, but their three dimensional structure or surface distribution of HA

peptide cannot be identical with wild type virus {McCraw, 2018 #485}, Thus, the HAI titers determined by VLPs flatform has to be differed from HAI titers determined by wild type viruses. Particularly in our study, we used VLPs as the immunization antigens. So the HAI titers were higher when using the same platform (VLPs) for the assay than using live virus (Fig 3). Thus, we applied ROC analysis to optimize the H7 VLP HAI titer cut-off to predict protection of antibodies elicited by H7 HA vaccinations [30]. The adjusted cut-off, 1:80 HAI unit, was more useful to predict protection against weight loss following Anhui/13 challenge than the 1:40 HAI titer. Our analysis based on optimized HAI cut-off for VLPs can be applied to predict protection efficacy of vaccines against multiple avian influenza variants, which could be difficult to obtain or propagate for animal challenge studies.

Serum HAI titer only reflects the protection mediated by the receptor binding antibodies. Influenza virus vaccines confer protection via diverse mechanisms, such as non-HAI antibodies or CD8+ cytotoxic T cells [12, 31]. Lung viral clearance may require multiple immune mechanisms, including antibodies, cytokines, dendritic cells and different T cell populations [32]. Blocking viral infection is known to be mediated by diverse mechanisms, such as neutralizing antibodies targeting non-receptor binding sites [33]. Until clear correlates of protection by non-HAI neutralizing antibodies or cell-mediated immune responses become available, the serum HAI titer will remain the most reliable indicator to evaluate influenza vaccine effectiveness.

One inherent limitation of this study was that the mouse model was used to extrapolate human antibody response to H7 HA immunization. Recent studies used ferrets as an alternative considering its high susceptibility to influenza virus, similar lung physiology and patterns of binding to sialic acid with human [34, 35]. Still, for the antibody research, ferret model might not be as useful considering that the ferret immunology has not well identified and there is no evidence that the ferret antibody can emulate the epitope recognition by human's. Rather, mouse model has advantages in antibody research, such as better availability, genetic homogenicty (inbred), and availability of diverse immunologic assay tools. Future study on broadly reactive H7 HA as a vaccine candidate should be evaluated for its efficacy in ferret challenge model.

In conclusion, the data presented in this study demonstrated that the cross reactive antibodies are elicited among H7 HA proteins, but the HA sequences are not correlated with the phylogenetic proximity or geographic orientation of the influenza HA antigens. Key amino acid mutations at putative antigenic sites in the H7 HA proteins are important for elicitation of broadly H7-reactive antibodies. Future studies will focus on developing vaccines to cover all known H7Nx influenza virus strains and future variants with key mutations.

## Supporting information

**S1 Fig. Determination of HAI cutoff using Receiver Operating Characteristic (ROC) curve analysis.** The plots of sensitivity% versus false positive rate (100-specificity%) of each cut-off were connected to form the ROC curve. Sensitivity = number of mice which showed hemagglutination inhibition (HAI) titer $\geq$ cut-off and was protected from the challenge study/all protected mice, Specificity = number of mice which showed hemagglutination inhibition (HAI) titer < cut-off and unprotected from the challenge study/ number of all unprotected mice, Youden's index = Sensitivity + Specificity -1.
(TIF)

**S2 Fig.**
(TIF)

**S3 Fig. HA A169T mutation resulted in slight increase in molecular weight to Anhui/13 VLPs.** The wild type (WT)-, and A169T mutant Anhui/13 VLPs were characterized by comassie blue stained SDS-PAGE. VLPs loaded on left two lanes were pre-treated with PNGaseF to remove the N-linked glycans. Red arrow indicates HA0 band for WT Anhui/13 VLPs.
(TIF)

**S1 Table. Sensitivity and specificity of 1:40 HAI cut off to predict protection defined by 90–100% of original body weight.**
(DOCX)

**S2 Table. Sensitivity and specificity of 1:80 HAI cut off to predict protection defined by 90–100% of original body weight.**
(DOCX)

**S3 Table. Sensitivity and specificity of 1:160 HAI cut off to predict protection defined by 90–100% of original body weight.**
(DOCX)

**S4 Table. Sensitivity and specificity of 1:320 HAI cut off to predict protection defined by 90–100% of original body weight.**
(DOCX)

## Acknowledgments

The authors would like to thank Amanada Skarlupka for technical assistance. Influenza viruses were provided by BEI resources (Manassas, VA, USA) and by Dr. Mark Tompkins (Athens, GA, USA) The authors would also like to thank the University of Georgia Animal Resource staff, technicians, and veterinarians for animal care and the staff of the Animal Health Research Center (AHRC) Biosafety Level 3 laboratories for providing biosafety and animal care. Also, the authors thank the members of the CVI protein production core, Jeffrey Ecker, Spencer Pierce, and Ethan Cooper, for providing technical assistance in purifying the recombinant proteins.

## Author Contributions

**Conceptualization:** Hyesun Jang, Ted M. Ross.

**Data curation:** Hyesun Jang.

**Formal analysis:** Hyesun Jang.

**Funding acquisition:** Ted M. Ross.

**Methodology:** Hyesun Jang.

**Project administration:** Ted M. Ross.

**Writing – original draft:** Hyesun Jang.

**Writing – review & editing:** Ted M. Ross.

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
