## [Decision Letter · Decision Letter 0]

12 Feb 2021

PONE-D-21-02219

Hemagglutination Inhibition (HAI) Antibody Landscapes after Vaccination with diverse H7 hemagglutinin (HA) proteins

PLOS ONE

Dear Dr. Ross,

Thank you for submitting your manuscript to PLOS ONE. After careful consideration, we feel that it has merit but does not fully meet PLOS ONE’s publication criteria as it currently stands. Therefore, we invite you to submit a revised version of the manuscript that addresses the points raised during the review process.

During the revision process, please address the reviewer comments related to data presentation and corrections within the text.

We look forward to receiving your revised manuscript.

Kind regards,

Victor C Huber

Academic Editor

PLOS ONE

Journal Requirements:

"All procedures were in accordance with the NRC Guide for Care and Use of Laboratory Animals, the Animal Welfare act, and the CDC/NIH Biosafety and Microbiological and Biomedical Laboratories (IACUC number A2017 11-021-Y3-A11).".   

Please amend your current ethics statement to confirm that your named ethics committee specifically approved this study.

For additional information about PLOS ONE submissions requirements for ethics oversight of animal work, please refer to http://journals.plos.org/plosone/s/submission-guidelines#loc-animal-research  

3.We note that the grant information you provided in the ‘Funding Information’ and ‘Financial Disclosure’ sections do not match.

Reviewers' comments:

Reviewer's Responses to Questions

**Comments to the Author**

1. Is the manuscript technically sound, and do the data support the conclusions?

Reviewer #1: Yes

Reviewer #2: Yes

2. Has the statistical analysis been performed appropriately and rigorously? 

Reviewer #1: No

Reviewer #2: Yes

3. Have the authors made all data underlying the findings in their manuscript fully available?

Reviewer #1: Yes

Reviewer #2: Yes

4. Is the manuscript presented in an intelligible fashion and written in standard English?

Reviewer #1: Yes

Reviewer #2: Yes

5. Review Comments to the Author

Reviewer #1: In their manuscript, Jang and Ross reported the results of mouse studies of a panel of H7 VLPs generated from H7 HA molecules of divergent influenza H7Nx viruses. The authors conducted in-depth evolutionary analysis of H7Nx viruses isolated over the two last decades and selected representative strains of different lineages to express VLPs exposing these HA molecules. Mice were immunized with these vaccine candidates to select the variant which induces most broadly-reactive antibodies. Challenge study was done with one H7N9 virus only; however since the vaccines induced various levels of the H7N9-reactive HAI antibodies, the authors had an opportunity to identify the cut-off HAI titer which correlate with protection. The mouse experiments demonstrated that some vaccine candidates induced only homologous antibodies, whereas others induced broadly reactive antibodies. This study is interesting and has an added value because provides scientific evidence for future choice of H7Nx virus for preparing candidate vaccine viruses for future H7 pandemic, should these viruses appear in circulation. The paper is well-written; however there are multiple grammar mistakes and typos which need to be corrected. Below are several comments that might further improve the quality of the manuscript.

1. The title refers to HA proteins, but the study assessed VLPs, not soluble proteins

2. Abstract and text say that a panel of nine HAs was chosen for VLP generation and mouse studies, whereas Table 1 and further figures demonstrate only eight strains. Please make the figures, tables and text consistent.

3. The details of determining the cut-off HAI titer needed for protection using ROC curve analysis between HAI titer and protection data are not clearly described. Were the individual data of weight loss for each mouse used for this analysis? Why the authors chose 95% of original body weigh as a measure of protection? It would be better to calculate the area under the curve of weight loss for each mouse and compare it with the corresponding HAI titer.

4. Page 2, lane 26 and other places: the term “receptor-blocking antibodies” is not correct – the antibodies block virus binding with receptors, but not receptors themselves.

5. Page 3, lane 15: add “5)” ahead of “”people in Europe”.

6. Page 4, lane 35. Study design should go to the Methods section.

7. Figure 1. A number of identified HA1 sequences should be indicated (not just “XX”)

8. Figure 2 doesn’t contain (A), (B), (C) signs.

9. Lane 207 – please indicate the color for this cluster on figure 2B.

10. Please correct Y axes titles for the figures 4: “HAI titer, log2” instead of “Log2 (HAI titers)” and figure 5c: “viral titer, log10PFU/ml” instead of “Log10(pfu/ml)”.

11. What specific human cell line was used for protein expression?

12. Page 5, lane 68. Is it really 5 to 10 grams of VLPs were coated?

13. P6, lane 90 earlist availibity

14. Lane 96 human endpoint

15. Lanes 89-94 – the protocol for mouse study mistakenly refers to Figure 2.

16. Lane 223-226 – please indicate that the amino acid differences and homologies refer to HA molecules.

17. Lane 230 – asteroid?

18. Lane 253: “HAI units”

19. Lane 274 – Youden’s indect

20. Lane 285 and other – number of mouse?

21. Figure 4 A,B – add significant differences if present. In addition, the figure caption is somewhat misleading. Detection of HA by antisera is not the same as blocking HA receptor binding by HAI antibodies. The figure shows HAI antibody titers against Anhui/13 H7 HA in mouse sera after immunization with various H7 HAs.

22. Figure 6. Also add statistical significance.

23. Lane 251: “… were tested…”.

24. There is no Mock-immunized mice on Figure 5, although they were mentioned in the main text (lanes 257-258). This group is essential for the figure 5C, where viral pulmonary titers are shown. The challenge virus was used at a high dose (100 LD), but infected mice in mock-immunized group should be still alive by day 4 post challenge.

Reviewer #2: The study investigates the antigenic difference between the H7 HA proteins derived from the circulating H7 influenza strains. Eight of H7 VLPs were generated and the HAI assay results demonstrated the cross reactivity of the VLP-induced antisera to against the antigens used. The authors show the site-specific glycosylation acts a role to affect the on the antigenicity, and determined the cut-off HAI titration that correlate to the protective immunity induced by the VLP. The study is well organized and only few questions to the manuscript.

1. Which human cell line was used for the production of VLP?

2. Page 5, line 67: “, A high affinity” changes to “, a high affinity”.

3. Page 5, line 68, “5 – 10 g of total protein”: Please check the amount of coating VLP for ELISA quantification assay.

4. Page 5, page 69. A recombinant H7 was generated in house, please add the reference in the content.

5. Page 6, line 92, 93 & 94, what is the data in Figure 2 associated with the immunization and viral examination?

6. Page 6, line 73: please note the source of the Ab (CR8020). If it was produce in the lab, please provide the reference.

7. Page 15, discussion section: Determination of HAI cut-off for protection

The HAI titration of the mice sera for each VLP was determined in the study, and the antisera with cut-off value of 1: 80 is correlated with the protective potency against the wild type influenza virus. Normally, the wild type influenza virus is used to measure HAI titer of the antisera instead of the VLP, while the study demonstrated there is a difference in the titration values to against the wild type virus and VLP. What is the factor cause the difference? And can you justify the effect of the different coating antigen in the determination of HAI cut-off value and its correlation with protective potency?

8. Page 16, line 307: A VLP of Anhui/13 A169 T was generated to investigate the cross reactivity of the induced Abs. As the site-specific glycan is the significant difference between these two viruses, I would suggest the authors add the characterization data in the article.

9. Page 16, Line 315, Fig. 6B changes to Fig. 7B.

6. PLOS authors have the option to publish the peer review history of their article (what does this mean?). If published, this will include your full peer review and any attached files.

Reviewer #1: No

Reviewer #2: No

---

## [Author Response · Author response to Decision Letter 0]

26 Feb 2021

5. Review Comments to the Author

Reviewer #1: In their manuscript, Jang and Ross reported the results of mouse studies of a panel of H7 VLPs generated from H7 HA molecules of divergent influenza H7Nx viruses. The authors conducted in-depth evolutionary analysis of H7Nx viruses isolated over the two last decades and selected representative strains of different lineages to express VLPs exposing these HA molecules. Mice were immunized with these vaccine candidates to select the variant which induces most broadly-reactive antibodies. Challenge study was done with one H7N9 virus only; however since the vaccines induced various levels of the H7N9-reactive HAI antibodies, the authors had an opportunity to identify the cut-off HAI titer which correlate with protection. The mouse experiments demonstrated that some vaccine candidates induced only homologous antibodies, whereas others induced broadly reactive antibodies. This study is interesting and has an added value because provides scientific evidence for future choice of H7Nx virus for preparing candidate vaccine viruses for future H7 pandemic, should these viruses appear in circulation. The paper is well-written; however there are multiple grammar mistakes and typos which need to be corrected. Below are several comments that might further improve the quality of the manuscript.

1. The title refers to HA proteins, but the study assessed VLPs, not soluble proteins

 Title was corrected 

2. Abstract and text say that a panel of nine HAs was chosen for VLP generation and mouse studies, whereas Table 1 and further figures demonstrate only eight strains. Please make the figures, tables and text consistent.

 Abstract was corrected

3. The details of determining the cut-off HAI titer needed for protection using ROC curve analysis between HAI titer and protection data are not clearly described. Were the individual data of weight loss for each mouse used for this analysis? Why the authors chose 95% of original body weigh as a measure of protection? It would be better to calculate the area under the curve of weight loss for each mouse and compare it with the corresponding HAI titer.

 1) Here in, we applied ROC curve analysis to determine HAI cut-off which can predict whether the ferrets were protected or not. The receiver operating characteristic curve (ROC curve) illustrates the diagnostic ability of “a binary classifier system” as its discrimination threshold is varied. 

Since we had to define the protection as in “the binary classifier”, we had to define the protection based on specific cut-off of %body weight. We agree with that to define protection solely from one arbitrary cut-off point might not be reasonable. So, instead of using one arbitrary measure (95% as a cut off to define protection), the definition of protection as % body weight 90-100% and added supplementary table 1-4 to show how the sensitivity and specificity changes based on the HAI cut-off and definition of body weight. The trend is still similar with our original analysis. We kept ROC curve based on 95% body weight to define the protection as a representative figure (Suppl. Fig. 1) 

2) We also agree with that our description for ROC curve analysis was not clear. A line was modified to clarify that we used individual % body weight to define protection. 

 3) Since the ROC curve analysis uses binary data, we have to define protection in a binary format. Thus, even though the area under the curve (AUC) is used to define protection, we will still need to use arbitrary cut-off point to define protection. The correlation assay between body weight AUC and HAI titer can provide the correlation two variables, but it won’t be to determine cut-off for serum HAI titer to predict protection.

 >Line 147-152, line 295-301, Suppl. Table. 1-4

4. Page 2, lane 26 and other places: the term “receptor-blocking antibodies” is not correct – the antibodies block virus binding with receptors, but not receptors themselves.

Corrected as receptor-binding antibodies

>Line 27 on p2, Line 118

5. Page 3, lane 15: add “5)” ahead of “”people in Europe”.

 Added as requested

 >Line 15 on p4

6. Page 4, lane 35. Study design should go to the Methods section.

 Changed as requested

 >Line 45-57 on p6

7. Figure 1. A number of identified HA1 sequences should be indicated (not just “XX”)

 Corrected as requested

8. Figure 2 doesn’t contain (A), (B), (C) signs.

Corrected as requested

9. Lane 207 – please indicate the color for this cluster on figure 2B.

 The color of pie for both North American H7N3 and H7N2 was indicated. Please note that the North American H7N2 resulted only one human infection case since 2013, so it was not able to be visualized in Fig. 2B 

 >Line 213-215 on p13

10. Please correct Y axes titles for the figures 4: “HAI titer, log2” instead of “Log2 (HAI titers)” and figure 5c: “viral titer, log10PFU/ml” instead of “Log10(pfu/ml)”.

Corrected as requested

11. What specific human cell line was used for protein expression?

 Cell information was added on Line 71-73, p7

12. Page 5, lane 68. Is it really 5 to 10 grams of VLPs were coated?

Corrected as µg on line 81, p7

13. P6, lane 90 earlist availibity

Corrected as requested (now line 103 on p8)

14. Lane 96 human endpoint

Corrected as requested (now line 109 on p8)

15. Lanes 89-94 – the protocol for mouse study mistakenly refers to Figure 2.

 Deleted (now line 102 on p8)

16. Lane 223-226 – please indicate that the amino acid differences and homologies refer to HA molecules.

Corrected as requested (now line 226-228 on p14)

17. Lane 230 – asteroid?

Corrected as asterisk (now line 252 on p15)

18. Lane 253: “HAI units”

Corrected as requested (now line 276 on p16)

19. Lane 274 – Youden’s indect

Corrected as index (now line 298 on p17)

20. Lane 285 and other – number of mouse?

Corrected “number of mice” (p.10, p17, p18)

21. Figure 4 A,B – add significant differences if present. In addition, the figure caption is somewhat misleading. Detection of HA by antisera is not the same as blocking HA receptor binding by HAI antibodies. The figure shows HAI antibody titers against Anhui/13 H7 HA in mouse sera after immunization with various H7 HAs.

The figure and caption was corrected as requested

22. Figure 6. Also add statistical significance.

The figure was corrected as requested

23. Lane 251: “… were tested…”.

 Corrected as requested (now line 274, p.16)

24. There is no Mock-immunized mice on Figure 5, although they were mentioned in the main text (lanes 257-258). This group is essential for the figure 5C, where viral pulmonary titers are shown. The challenge virus was used at a high dose (100 LD), but infected mice in mock-immunized group should be still alive by day 4 post challenge.

Mock group was added as requested

Reviewer #2: The study investigates the antigenic difference between the H7 HA proteins derived from the circulating H7 influenza strains. Eight of H7 VLPs were generated and the HAI assay results demonstrated the cross reactivity of the VLP-induced antisera to against the antigens used. The authors show the site-specific glycosylation acts a role to affect the on the antigenicity, and determined the cut-off HAI titration that correlate to the protective immunity induced by the VLP. The study is well organized and only few questions to the manuscript.

1. Which human cell line was used for the production of VLP?

 Cell information was added on Line 71-73, p7

2. Page 5, line 67: “, A high affinity” changes to “, a high affinity”.

Corrected as requested (now on Line 80, p7)

3. Page 5, line 68, “5 – 10 g of total protein”: Please check the amount of coating VLP for ELISA quantification assay.

Corrected as µg on line 81, p7

4. Page 5, page 69. A recombinant H7 was generated in house, please add the reference in the content.

 Reference was added (line 82, p7)

5. Page 6, line 92, 93 & 94, what is the data in Figure 2 associated with the immunization and viral examination?

The word “Figure 2” was deleted (now line 102, p.8)

6. Page 6, line 73: please note the source of the Ab (CR8020). If it was produce in the lab, please provide the reference.

Reference was added (now line 86, p7)

7. Page 15, discussion section: Determination of HAI cut-off for protection

The HAI titration of the mice sera for each VLP was determined in the study, and the antisera with cut-off value of 1: 80 is correlated with the protective potency against the wild type influenza virus. Normally, the wild type influenza virus is used to measure HAI titer of the antisera instead of the VLP, while the study demonstrated there is a difference in the titration values to against the wild type virus and VLP. What is the factor cause the difference? And can you justify the effect of the different coating antigen in the determination of HAI cut-off value and its correlation with protective potency?

 Despite that the VLPs carry the same HA amino acid sequence with wild type virus, the VLPs are differed in multiple characteristics, such as the three dimensional structure and expression pattern/ratio of HAs and NAs on the surface. Particularly for this study, we used VLPs for immunization so it was not surprising that we observed higher HAI titer to H7 VLPs than WT Anhui/13 H7N9 virus. 

The HAI titer has been used as surrogate for the protection, especially to prove the efficacy of influenza vaccine candidates. We acknowledge that conducting challenge study with wild type virus is the most ideal to prove the efficacy of vaccine. But we wanted to develop an alternative platform to predict protective efficacy against H7 variant viruses. Having such platform can significantly accelerate response to emergence of novel H7 variants, since expression of VLPs is much faster than propagation of novel variants.

We modified the paragraph (line 461-478, p.27) to address this point.

8. Page 16, line 307: A VLP of Anhui/13 A169 T was generated to investigate the cross reactivity of the induced Abs. As the site-specific glycan is the significant difference between these two viruses, I would suggest the authors add the characterization data in the article.

 We added comassie blue stained SDS-PAGE results as Suppl. Fig. 3 as characterization data (M&M on p.11, results on p.23).

9. Page 16, Line 315, Fig. 6B changes to Fig. 7B.

Corrected as requested (now on line 352, p.23)

---

## [Editor Report · Decision Letter 1]

2 Mar 2021

Hemagglutination Inhibition (HAI) Antibody Landscapes after Vaccination with H7Nx Virus Like Particles

PONE-D-21-02219R1

Dear Dr. Ross,

We’re pleased to inform you that your manuscript has been judged scientifically suitable for publication and will be formally accepted for publication once it meets all outstanding technical requirements.

Kind regards,

Victor C Huber

Academic Editor

PLOS ONE
---

## [Editor Report · Acceptance letter]

8 Mar 2021

PONE-D-21-02219R1 

Hemagglutination Inhibition (HAI) Antibody Landscapes after Vaccination with H7Nx Virus Like Particles 

Dear Dr. Ross:

I'm pleased to inform you that your manuscript has been deemed suitable for publication in PLOS ONE. Congratulations! Your manuscript is now with our production department. 

Kind regards, 

on behalf of

Dr. Victor C Huber 

Academic Editor

PLOS ONE